# The influence of leprosy-related clinical and epidemiological variables in the occurrence and severity of COVID-19: A prospective real-world cohort study

**Selma Regina Penha Silva Cerqueira**[1,2], **Patrícia Duarte Deps**[3], **Débora Vilela Cunha**[2], **Natanael Victor Furtunato Bezerra**[1], **Daniel Holanda Barroso**[1,2], **Ana Bárbara Sapienza Pinheiro**[4], **Gecilmara Salviato Pillegi**[5], **Taynah Alves Rocha Repsold**[3], **Patrícia Shu Kurizky**[1,2], **Simon M. Collin**[3], **Ciro Martins Gomes**[1,2,4] *

**1** Programa de Pós-Graduação em Ciências Médicas, Faculdade de Medicina, Universidade de Brasília, Brasília, Brazil, **2** Hospital Universitário de Brasília, Universidade de Brasília, Brasília, Brazil, **3** Post-Graduation Programme of Infectious Diseases, Federal University of Espírito Santo, Vitória, Brazil, **4** Programa de Pós-Graduação em Medicina Tropical, Faculdade de Medicina, Universidade de Brasília, Brasília, Brazil, **5** Disciplina de Reumatologia da Universidade Federal de São Paulo/EPM, São Paulo, Brazil

* ciromgomes@gmail.com

## Abstract

### Background

Protective effects of Bacillus Calmette–Guérin (BCG) vaccination and clofazimine and dapsone treatment against severe acute respiratory syndrome coronavirus 2 (SARS-CoV-2) infection have been reported. Patients at risk for leprosy represent an interesting model for assessing the effects of these therapies on the occurrence and severity of coronavirus disease 2019 (COVID-19). We assessed the influence of leprosy-related variables in the occurrence and severity of COVID-19.

### Methodology/Principal findings

We performed a 14-month prospective real-world cohort study in which the main risk factor was 2 previous vaccinations with BCG and the main outcome was COVID-19 detection by reverse transcription polymerase chain reaction (RT-PCR). A Cox proportional hazards model was used. Among the 406 included patients, 113 were diagnosed with leprosy. During follow-up, 69 (16.99%) patients contracted COVID-19. Survival analysis showed that leprosy was associated with COVID-19 (p<0.001), but multivariate analysis showed that only COVID-19-positive household contacts (hazard ratio (HR) = 8.04; 95% CI = 4.93–13.11) and diabetes mellitus (HR = 2.06; 95% CI = 1.04–4.06) were significant risk factors for COVID-19.

### Conclusions/Significance

Leprosy patients are vulnerable to COVID-19 because they have more frequent contact with SARS-CoV-2-infected patients, possibly due to social and economic limitations. Our

**Data Availability Statement:** All relevant data are within the manuscript and its Supporting Information files.

**Funding:** This study was financed in part by the Coordenação de Aperfeiçoamento de Pessoal de Nível Superior - Brasil (CAPES) - Finance Code 001 to CMG. The funders had no role in study design, data collection and analysis, decision to publish, or preparation of the manuscript.

**Competing interests:** The authors have declared that no competing interests exist.

model showed that the use of corticosteroids, thalidomide, pentoxifylline, clofazimine, or dapsone or BCG vaccination did not affect the occurrence or severity of COVID-19.

## Author summary

Leprosy is a chronic infectious disease caused by *Mycobacterium leprae* and *Mycobacterium lepromatosis*. Coronavirus disease 2019 (COVID-19) results in an intense inflammatory response that is dependent on several immune mediators. Since the beginning of the COVID-19 pandemic, an interaction of severe acute respiratory syndrome coronavirus 2 (SARS-CoV-2) and *M. leprae* infection has been considered a possibility. Protective effects of Bacillus Calmette–Guérin (BCG) vaccination and clofazimine and dapsone treatment against SARS-CoV-2 infection have been reported. Patients at risk for leprosy represent an interesting model for assessing the effects of these therapies on the occurrence and severity of COVID-19. We assessed the influence of leprosy-related variables in the occurrence and severity of COVID-19. We performed a 14-month study. Among the 406 included patients, 113 were diagnosed with leprosy. During follow-up, 69 (16.99%) patients contracted COVID-19. Our analysis showed that only COVID-19-positive household contacts and diabetes mellitus were significant risk factors for COVID-19. Leprosy patients are vulnerable to COVID-19 because they have more frequent contact with SARS-CoV-2-infected patients, possibly due to social and economic limitations. Our model showed that the use of clofazimine or dapsone or BCG vaccination did not affect the occurrence or severity of COVID-19.

## Introduction

Leprosy is a chronic infectious disease caused by *Mycobacterium leprae* and *Mycobacterium lepromatosis*.[1] Affected patients may experience a variety of neurological and cutaneous symptoms because of the immunological response stimulated by the bacillus.[2] Since the beginning of the coronavirus disease 2019 (COVID-19) pandemic, an interaction of severe acute respiratory syndrome coronavirus 2 (SARS-CoV-2) and *M. leprae* infection has been considered a possibility.[3] Preliminary data suggested that coinfection could result in a higher incidence of leprosy reactions.[4]

Leprosy reactions are inflammatory conditions that are triggered by diverse factors, including bacillus destruction and viral infection.[5] Most patients infected by *M. leprae* do not develop clinical disease, but the clinical presentation of the disease is clearly dependent on the Th1/Th2 immune response.[6] Leprosy patients who initiate a Th1-polarized cellular-based immune response develop limited skin lesions, with few bacilli. Immune mediators, such as tumour necrosis factor (TNF), interferon-γ (IFN-γ), interleukin-6 (IL-6) and IL-12, are essential for disease control but are also responsible for neural damage.[4] In contrast, patients who initiate a humoral Th2-based response develop diffuse and infiltrative diseases. Patients who initiate immune responses between these extremes may have a variety of clinical presentations, which are referred to as borderline forms.[7]

COVID-19 results in an intense inflammatory response that is dependent on several mediators, including TNF and IL-6.[8] Cytokine storm can result in severe pulmonary inflammation.[9] Many attempts to modulate this erratic immune response have been described, but these have shown little success.[10,11] A recent publication reported that SARS-CoV-2/*M.*

*leprae* coinfection was followed by intense IL-6- and IL-12-dependent responses, but the clinical presentations of both diseases were not changed.[4]

Epidemiological differences in the number of cases of COVID-19 suggested a protective role of the innate Th1 immune response. Although recent studies suggest no relationship between neonatal application of Bacillus Calmette-Guérin (BCG) and COVID-19,[12] a possible protective cross-immune reaction by mycobacterial infection and BCG against SARS-CoV-2 infection has been proposed.[13] There are no consistent data related to BCG revaccination and COVID-19. Clofazimine and dapsone, which are used for the treatment of leprosy, also showed protective effects against SARS-CoV-2 infection in laboratory models.[14–16] Because leprosy is a mycobacterial infection that is highly dependent on Th1/Th2 modulation, and leprosy prophylaxis involves BCG vaccination in some countries, leprosy-endemic regions are interesting settings for assessments of the effects of these variables on the occurrence of COVID-19 in leprosy patients.

The main objective of this study was to assess the influence of leprosy-related clinical and epidemiological variables as risk/protective factors for the occurrence and severity of COVID-19. We also examined the influence of the number of BCG doses and exposure to *M. leprae* infection on the occurrence of COVID-19. Finally, we evaluated the association of the use of medications for leprosy and leprosy reaction with the risk of COVID-19.

## Methods

### Ethics statement

The study complied with the Declaration of Helsinki (2013 revision). Participants were included after signing an informed consent form. For child participants formal consent was obtained from the parent/guardian. The Ethics Committee of the Faculty of Medicine, Universidade de Brasília approved the study (34164820.6.0000.0030). This study also complied with The Strengthening the Reporting of Observational Studies in Epidemiology (STROBE) statement (S1 STROBE Checklist).[17] We performed a 14-month prospective real-world cohort study in which, for better statistical follow-up, the main exposure was defined as 2 previous BCG vaccinations, and the main outcome was defined as COVID-19 detection by reverse transcription–polymerase chain reaction (RT-PCR). BCG status (0, 1 or 2 doses) and leprosy exposure status (active disease, household contact of a patient with leprosy or control) were defined as secondary exposures of interest. Time at risk was calculated from data enrolment to the end of follow-up, censoring or the onset of COVID-19.

We chose the abovementioned primary and secondary exposures of interest to assess the most important leprosy-related variables because these factors are closely related to leprosy epidemiology. A first dose of BCG is indicated at birth for all Brazilians, and a second dose, although mandatory for schoolchildren through the end of the 20th century, is currently administered only to household contacts of leprosy patients.[18] Based on previous data from the recruitment site, we recruited a prospective cohort comprising three main groups of approximately equal size: 1) current (active) leprosy patients; 2) household contacts (HHCs) of these leprosy patients; and 3) dermatology outpatient clinic controls (individuals assessed for leprosy but with differential diagnosis).

From March to May 2020, patients were enrolled consecutively at the Leprosy Outpatient Service of Brasília University Hospital, University of Brasília, Brazil. This site supports more than 60% of all leprosy patients in the region, which has a population of more than 3 million people, and it performs differential diagnoses in all cases referred from secondary and primary healthcare facilities. The inclusion criteria consisted of all patients referred to the abovementioned outpatient service, which is a population formed by patients with active leprosy,

household contacts of patients with leprosy and individuals with clinical conditions with a differential diagnosis of leprosy (controls). We excluded patients with a previous diagnosis of COVID-19 and any use of immunosuppressive drugs, except drugs that were used to treat leprosy reactional states. Vaccination against COVID-19 was not available when the study began, and vaccinated patients were subsequently censored from follow-up to avoid any effect of this specific immunization on the outcomes.

At the first interview, patients were invited to participate in the study, and clinical information was collected. Patients were evaluated monthly at face-to-face medical consultations, via assessments of electronic medical files or telephone to collect information related to the target outcomes, including COVID-19 occurrence and the presence of complications.

A diagnosis of leprosy was made according to the World Health Organization (WHO) criteria,[19] and any patient receiving multidrug therapy was considered an active leprosy case. Any patient who reported leprosy and had already completed treatment was considered to have previous leprosy. Any patient suspected of having type I or type II leprosy reactions was evaluated by a dermatologist monthly or more frequently. A leprosy HHC was defined as any person who lived with a leprosy patient within the last 5 years.[18] The WHO disability grading system was used for physical impairment classification.[20] Leprosy classification followed the Ridley and Jopling criteria. A COVID-19 HHC was defined as any patient who reported a previous HHC with a person with a positive RT-PCR test for SARS-CoV-2.

## Confounders

Variables assessed as possible confounders were age, sex, obesity, hypertension, diabetes, smoking, alcohol abuse, drug abuse, previous leprosy, full compliance with personal protection measures against SARS-CoV-2 infection (e.g., face mask, alcohol gel, and frequent hand washing), adoption of social distancing behaviours, and HHC with a patient with confirmed COVID-19. Personal protection against COVID-19 was defined according to the following criteria: 1. Adequate personal protection: Patients reported having received adequate orientation and acting in full compliance with recommendations related to the use of masks in public environments, alcohol gel and frequent hand washing; 2. Social distancing: Full compliance with local recommendations and local liberations and a complete lack of participation in labour or leisure activities not allowed by current local decrees. Clinical variables related to leprosy treatment (clofazimine use, dapsone use, rifampicin use, corticosteroid use, thalidomide use, and pentoxifylline use) were also analysed in the subgroup of active leprosy patients. Additional variables related to ethnic, social and economic conditions were also investigated. Different ethnic groups were classified according to the Brazilian Institute of Geography and Statistics (IBGE) as reported by the individual evaluated. Unemployment was defined when the patient had no formal or informal source of personal income.

## Alternative outcomes

Additional outcomes consisted of the severity of COVID-19. Severe COVID-19 was defined as the necessity for $O_2$ supplementation, mechanical ventilation, or intensive care support or as death. We additionally analysed the association of SARS-CoV-2/*M. leprae* coinfection with the occurrence of leprosy reactions and disabilities.

## Statistical analysis

For the main exposure, secondary exposure and possible confounders, unadjusted relative risks (RRs) and hazard ratios (HRs) were calculated, and log-rank tests and survival evaluations by the Kaplan-Meier method were performed to detect any possible influence on the

occurrence of COVID-19. Missing data, although rare, were removed from the statistical analysis and considered a negative result for crude frequency and percentage calculations.

For the main exposure, a hierarchical model in which the occurrence of COVID-19 was defined as the main outcome was constructed. The model was formed by independent variables considered clinically relevant for the occurrence of COVID-19. Two dermatology specialists (CG & PK) made this decision. The following variables were included in a three-block model in ascending order of relevance for the outcome: 1. sex, age, hypertension, and obesity; 2. diabetes, drug abuse, and thalidomide use; and 3. corticosteroid, clofazimine, and dapsone use, a leprosy diagnosis, a COVID-19-positive HHC, and the number of BCG doses received. For the main exposure and secondary exposures, we calculated HRs using a Cox proportional hazards model, with proportionality tested using Schoenfeld residuals supported by visual inspection of log-log plots

For all secondary exposures, we used an exploratory strategy. We used subgroup analysis to evaluate only leprosy patients. Risk factors that met a low evidential threshold (p≤0.1) for possible associations with leprosy status and COVID-19 and that were not deemed to be on a causal pathway were carried forward to a 'partially adjusted' multivariable model. A 'fully adjusted' multivariable model that included all of the measured risk factors was fitted in a sensitivity analysis to assess the extent of residual confounding. An a priori hypothetical interaction between leprosy status and BCG status was tested in both models using likelihood ratio tests.

We used the survival (Therneau T (2021). _A Package for Survival Analysis in R_. R package version 3.2–10, <URL: https://CRAN.R-project.org/package=survival>.) and survminer (Alboukadel Kassambara, Marcin Kosinski and Przemyslaw Biecek (2021). survminer: Drawing Survival Curves using 'ggplot2'. R package version 0.4.9. https://CRAN.R-project.org/package=survminer) packages in R Studio (R Studio: Integrated Development for R. R Studio, PBC, Boston, MA URL http://www.rstudio.com/.), based on R Core Team (2020). R: A language and environment for statistical computing. R Foundation for Statistical Computing, Vienna, Austria. URL https://www.R-project.org/. Statistical significance was defined by a p value <0.05 and an appropriate 95% confidence interval (CI).

### Sample size

Our sample size calculation was limited by the scarcity of data comparing leprosy-related variables with the occurrence of COVID-19. Before recruitment, we considered that the only variable suitable for sample size calculation was BCG vaccination [13]. This variable would warrant external validation and generalizability related to BCG vaccination and the occurrence of COVID-19 according to the power and probability used in the calculation as follows. We arbitrarily considered that 50% of patients with fewer than two BCG vaccinations would develop COVID-19 and that only 35% of patients with two vaccinations would develop symptomatic SARS-CoV-2 infection (1-alpha = 95, 1-beta = 80%; relative size cases/controls = 1). [21] Our sample size calculation resulted in a minimum of 405 patients, which already included a 10% buffer added to the intended population size, to avoid the effects of loss. The remaining associations were evaluated in the cohort of patients included in the study warranting internal validation. External validation of the remaining variables was performed using an exploratory strategy with post hoc analysis.

### Results

We included 406 individuals, and during the follow-up period, 69 (16.99%) of these individuals developed RT-PCR-confirmed COVID-19 (Table 1 and Fig 1 and S1 Table). A total of 113 patients had active leprosy, 153 individuals were leprosy HHCs, and 140 individuals were classified as controls. The active leprosy patients were classified as follows: 1 patient had

**Table 1. COVID-19 exposures and outcomes in the cohort (n = 406).**

| | Control (n = 140) | Leprosy HHC (n = 153) | Active leprosy (n = 113) | P-value† |
|---|---|---|---|---|
| **COVID-19 exposures** | | | | |
| Personal protection | 125 (89.29%) | 147 (96.08%) | 101 (89.38%) | 0.054 |
| Social distancing | 134 (95.71%) | 106 (69.28%) | 94 (83.19%) | <0.001 |
| Household case | 9 (6.43%) | 37 (24.18%) | 29 (25.66%) | <0.001 |
| **COVID-19 outcomes** | | | | |
| Confirmed diagnosis | 10 (7.14%) | 26 (16.99%) | 33 (29.20%) | <0.001 |
| | (n = 10) | (n = 26) | (n = 33) | |
| Time to onset (months) | 7.80 (3.22) | 6.73 (3.35) | 7.30 (3.21) | 0.655 |
| Duration of illness (days) | 8.30 (5.76) | 11.19 (11.61) | 11.73 (4.84) | 0.262 |
| Hospital emergency admission | 4 (40.00%) | 4 (15.38%) | 8 (24.24%) | 0.282 |
| ICU admission | 2 (20.00%) | 1 (3.84%) | 4 (12.12%) | 0.238 |
| Supplemental $O_2$ | 1 (10.00%) | 1 (3.85%) | 4 (12.12%) | 0.508 |
| Mechanical ventilation | 0 (0.0%) | 0 (0.0%) | 0 (0.0%) | - |
| Death | 0 (0.0%) | 0 (0.0%) | 0 (0.0%) | - |

The table shows the frequency (%) or mean (standard deviation); n = number of patients; HHC = household contact; ICU = intensive care unit

† Chi-squared test for categorical variables (or Fisher's exact test if frequency ≤ 5); one-way ANOVA or Kruskal-Wallis tests for continuous variables

indeterminate leprosy; 12 patients had tuberculoid-tuberculoid leprosy; 20 patients had tuberculoid-borderline leprosy; 35 patients had borderline-borderline leprosy; 26 patients had borderline-lepromatous leprosy; and 19 patients had lepromatous leprosy. The control group comprised 93 individuals with localized allergic dermatosis (nummular eczema, contact dermatitis and pityriasis alba), 12 individuals with superficial mycosis and 35 individuals with neuropathy of axial or traumatic origin. The frequency of confirmed COVID-19 cases was higher in leprosy patients, with 33 cases (29.20%) in active leprosy patients, 26 cases (16.99%) in leprosy HHCs, and 10 cases (7.14%) in controls (Table 1 and Fig 1).

## Main exposure

Patients with leprosy had a higher risk of developing COVID-19 than patients without leprosy in our unadjusted survival analysis (S1 File), However, the hierarchical model showed that this

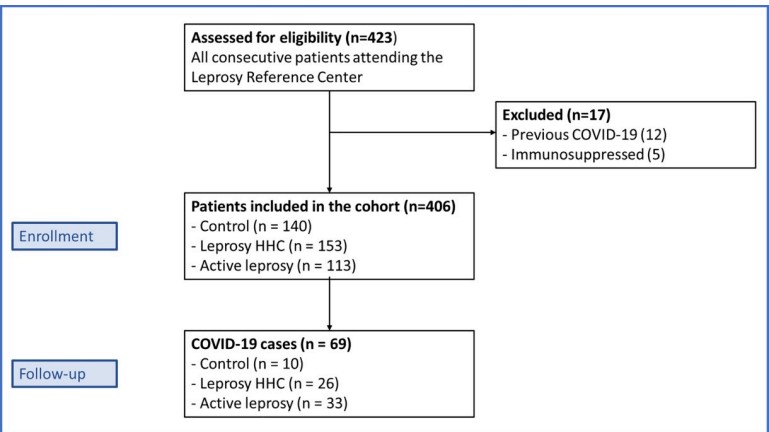

**Fig 1. Study flow chart.** HHC = household contact.

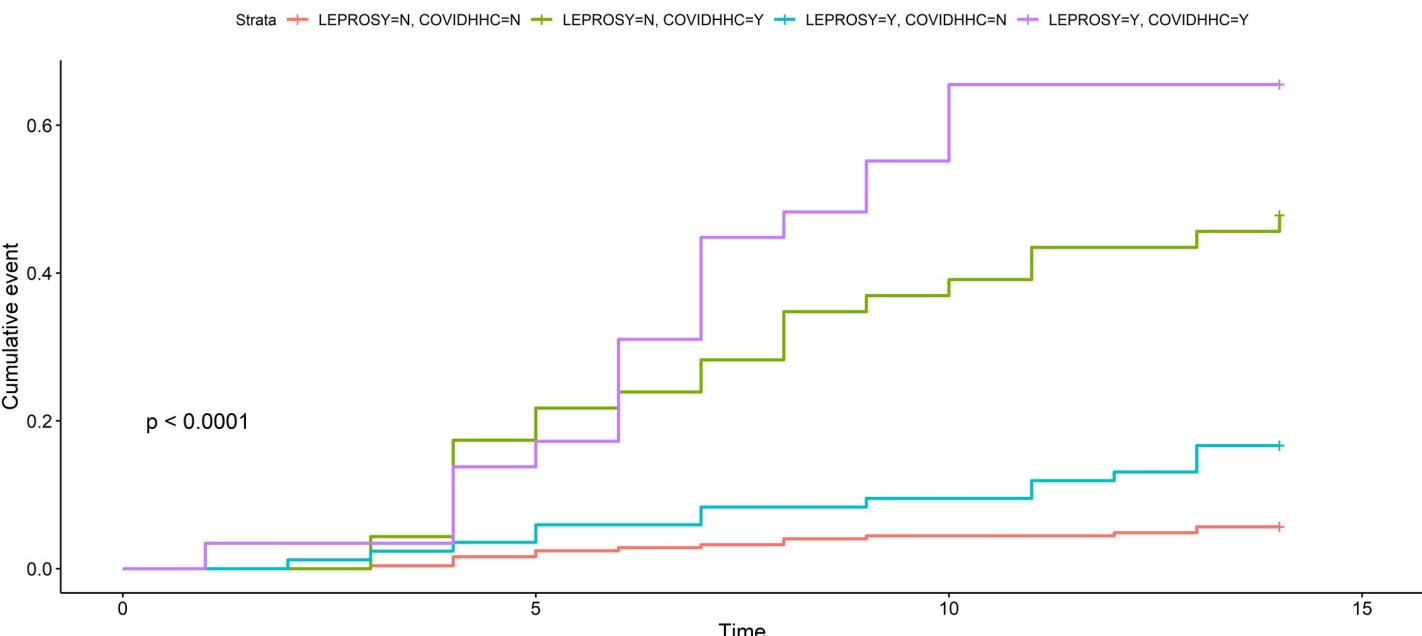

**Fig 2. Survival curves adjusted for active leprosy and COVID-19 positive household contact representing possible risk factors for the occurrence of COVID-19.** N = no, Y = yes, HHC = household contacts.

risk was probably related to more frequent exposure to HHCs infected with SARS-CoV-2. Leprosy patients were more frequently exposed to SARS-CoV-2-infected HHCs (n = 33/113 (29.20%)) than non-leprosy patients (leprosy HHCs + controls; n = 36/293 (12.29%)) (p<0.001). When the COVID-19 HR was analysed in leprosy versus non-leprosy patients (leprosy HHCs + controls) and the results were adjusted only for COVID-19 HHC history, we found that leprosy still influenced the occurrence of this viral infection (HR = 2.08; 95% CI = 1.29–3.36)(Fig 2). Our predefined hierarchical Cox proportional hazards model showed that a positive history of a COVID-19-positive HHC was a significant risk factor for the occurrence of COVID-19 (HR = 8.04; 95% CI = 4.93–13.11). Diabetes mellitus also enhanced the risk of COVID-19 occurrence (HR = 2.06; 95% CI = 1.04–4.06)(Fig 3). This principal statistical model showed that sex, age, hypertension, obesity, drug abuse, thalidomide use, corticosteroid use, clofazimine use, dapsone use, active leprosy, a COVID-19-positive HHC, and the number of BCG doses received were neither risk nor protective factors for the occurrence of COVID-19 (Fig 3).

## Exploratory analysis

In unadjusted analysis with the control group as the reference group, the corresponding HRs for COVID-19 occurrence were 2.55 (95% CI = 1.23–5.28) for the leprosy HHC group and 4.62 (95% CI = 2.28–9.38) for the active leprosy group (Table 2). Only active leprosy patients presented a higher HR in an exploratory fully adjusted model (HR = 2.91; 95% CI = 1.33–6.32) (Table 2). This fully adjusted exploratory analysis, similar to our principal model, showed no evidence for protective effects of 1 dose or 2 doses of BCG compared with 0 doses. COVID-19 HHC increased the risk of COVID-19 more than 7-fold (HR = 7.43 (95% CI = 4.35–12.68)) (Table 2).

Most variables related to social conditions were exceedingly difficult to represent in this study due to the subjectivity of the data. However, unemployment affected 47 active leprosy

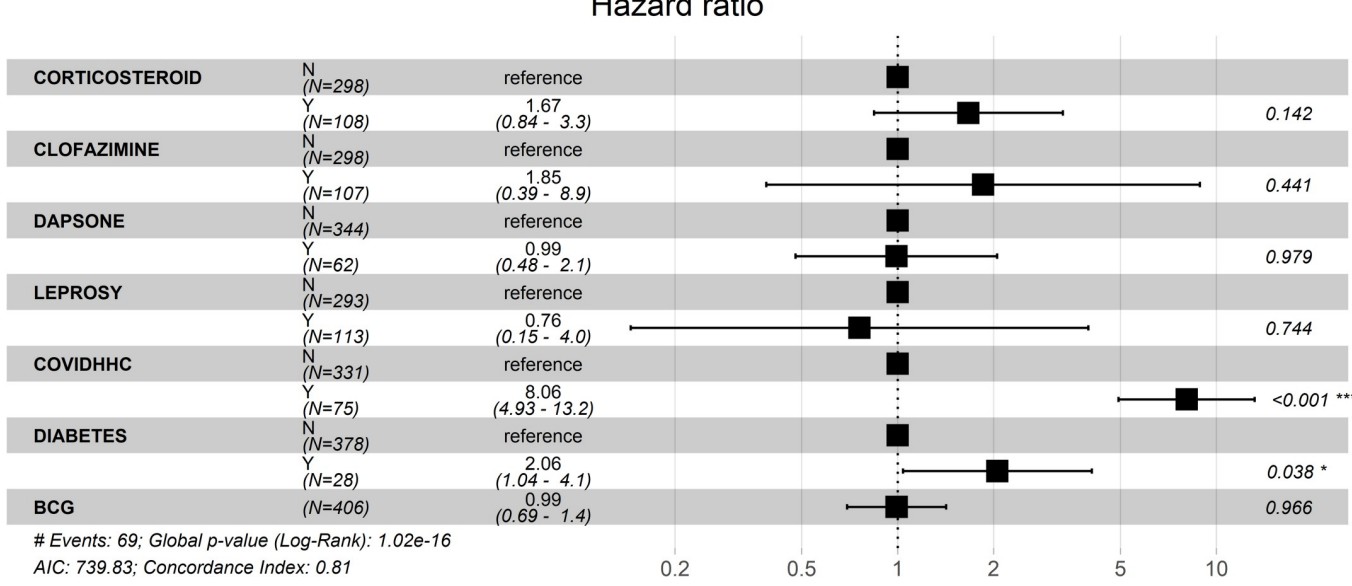

**Fig 3. Hazard ratios and 95% confidence intervals calculated by the Cox proportional hazards model.**

patients (41,59%), 37 leprosy HHCs (24.18%) and 20 controls (14.29%). The unemployment rate of active leprosy patients was higher than that of HHCs (p = 0.003) and controls (p<0.001). The unemployment rate of leprosy HHCs was also higher than that of controls (p = 0.047). The ethnic distribution of the groups was also different. Active leprosy patients were more commonly members of black (31.97%) and brown (33.63%) ethnic groups than leprosy HHCs (p = 0.003) and controls (p = 0.007). In contrast, 52.14% of controls, 45.10% of leprosy HHCs and 33.63% of controls were classified as white (S1 File).

## Alternative outcomes–COVID-19 severity and the presence of physical disabilities

No patient died from COVID-19 in our total sample. Age was the only factor associated with severe outcomes in patients diagnosed with COVID-19 (p = 0.005) (S1 File). SARS-CoV-2

**Table 2. Exploratory analysis testing the effects of leprosy status, BCG and COVID household contact on the risk of COVID-19 (N = 406).**

| | | Confirmed COVID-19 | Unadjusted HR (95% CI) | Fully adjusted HR (95% CI) † |
|---|---|---|---|---|
| Leprosy status | Control (n = 140) | 10 | 1.00 | 1.00 |
| | Leprosy HHC (n = 153) | 26 | 2.55 (1.23–5.28) p = 0.012 | 2.22 (0.90–5.43) p = 0.080 |
| | Active leprosy (n = 113) | 33 | 4.62 (2.28–9.38) p < 0.001 | 2.91 (1.33–6.32) 0.007 |
| BCG vaccination (doses) | 0 (n = 71) | 18 | 1.00 | 1.00 |
| | 1 (n = 207) | 32 | 0.56 (0.31–0.99) p = 0.048 | 0.98 (0.52–1.85) p = 0.956 |
| | 2 (n = 128) | 19 | 0.54 (0.28–1.02) p = 0.058 | 0.63 (0.28–1.39) p = 0.259 |
| COVID-19 household contact | No (n = 331) | 28 | 1.00 | 1.00 |
| | Yes (n = 75) | 41 | 8.96 (5.53–14.54) p = <0.001 | 7.43 (4.35–12.68) p < 0.001 |

n = number of patients; HR = hazard ratio; CI = confidence interval; HHC = household contact; BCG = Bacillus Calmette-Guérin

† Adjusted for age group, sex, obesity, hypertension, diabetes, smoking, alcohol abuse, drug abuse, personal protective equipment use, social distancing, previous leprosy, BCG doses, and COVID-19 HHCs. In the fully adjusted analysis, no statistical cut-off was used for variable selection.

infection was not associated with the frequency or intensity of leprosy reactions. Additionally, there was no association between disability grade and COVID-19 occurrence (S1 File).

## Discussion

Since the beginning of the COVID-19 pandemic, the evolution of SARS-CoV-2/*M. leprae* coinfection has attracted great interest. This interest is justified by the possibility of refractory leprosy reactional states during the inflammatory phase of COVID-19.[22] Laboratory studies suggested possible suppression of SARS-CoV-2 in patients with a stimulated cellular immunological response, a state that may be achieved after BCG vaccination.[13] The present report is the first large prospective study based on primary data collected from a leprosy-endemic region to assess risk factors for the development of COVID-19.

A previous descriptive study performed in northeastern Brazil showed that only 1% of 378 leprosy patients developed COVID-19. Although this percentage may represent possible protection against COVID-19, we must also consider that this survey was performed during the first epidemic peak, and the rate of COVID-19 has more than doubled in this region.[23,24] A cross-sectional study performed in midwestern Brazil after the first epidemic peak of COVID-19 showed that 18.75% of leprosy patients developed viral infection.[4] The initial studies were likely biased by the phase of the pandemic and the local infection rate. Our study showed a COVID-19 incidence of 29.20% in leprosy patients.

The present survival analysis showed that active leprosy patients were more frequently affected by COVID-19 than leprosy HHCs and controls. Although confounding factors likely influenced this association, this result may support the hypothesis that relative immunosuppression in leprosy could increase susceptibility to SARS-CoV-2.[25] If this assumption is true, multibacillary patients may have a higher risk of acquiring COVID-19. However, our sample of active leprosy patients was not adequate for testing this hypothesis. To avoid confounding factors, we used a previously designed multivariate Cox proportional hazards model. The principal model showed that no characteristic related to leprosy or BCG vaccination acted as a risk factor for or protective factor against COVID-19. Alternatively, this model showed that previous histories of HHC with COVID-19 and diabetes were risk factors for COVID-19. Although we may assume that leprosy is not a direct risk factor for COVID-19, the clinical relevance of our survival analysis must not be neglected (Fig 2), especially because a post hoc analysis of the influence of leprosy on COVID-19 showed a very narrow 95% CI, ranging from 1.62 to 4.16.

Leprosy patients were more frequently exposed to SARS-CoV-2-infected HHC than non-leprosy patients. This result reinforces the possibility that leprosy patients are more vulnerable to COVID-19 due to other social or health-related factors. It is widely known that poverty is associated with leprosy.[26] Socio-economic deprivation also influences the occurrence of COVID-19. Families with economic difficulties are more prone to violate social distancing recommendations to provide basic needs for families. Although the COVID-19 pandemic initially affected developed countries, there is growing evidence that poverty is related to a greater risk of COVID-19.[26,27] Some variables reinforce this possibility. The unemployment rate was considered high in the studied population, especially in leprosy patients. Ethnic distribution of patients also represented Brazilian social and economic inequalities because brown and black ethnic groups are generally more vulnerable to leprosy. However, other social and economic factors likely play an important role in the risk of developing COVID-19, and these factors must be better analysed in future studies designed for this purpose, including qualitative analysis.

The only risk factor for a severe outcome of COVID-19 infection in the total population was age (S1 File). Elderly patients probably compose a population that is more frequently

affected by severe COVID-19. Age is also related to intense pulmonary inflammation.[28] Other risk factors, such as obesity and hypertension, were not related to severe COVID-19. However, we must consider that the population size was not sufficient to detect these secondary outcomes. Contrary to the previous assumption, BCG vaccination was not related to COVID-19 severity. This result is consistent with recent reports.[29]

Consistent with the results of a previous cross-sectional study, leprosy/COVID-19 coinfection was not related to a higher frequency of Type I or Type II leprosy reactions. Coinfected patients had no more morbidities than leprosy patients without COVID-19 (S1 File). Although COVID-19 apparently did not influence leprosy reactions, we must not neglect the potential for a chronic inflammatory state to induce neural damage. A previous study showed that IL-6 and IL-12B inflammatory mediators were chronically elevated in leprosy patients who developed COVID-19.[4] Leprosy patients represent a vulnerable population that needs attention. [30,31]

Although the intended sample size was achieved, and the final model showed a clinically relevant result, some limitations must be acknowledged. The existence of confounders must always be considered in observational protocols, although prospective studies are paramount for adequate clinical trial design. New trials are important to test the effects of drugs, such as clofazimine and dapsone, on COVID-19 development. A previous *in vitro* study reported antiviral properties of clofazimine at a 200 mg daily dose.[14–16] The usual dose for the treatment of leprosy, 50 mg, may not have a suppressive effect on SARS-CoV-2. No patient used a higher dose of clofazimine.[32]

We conclude that leprosy patients may be vulnerable to COVID-19, although immunological factors are apparently not involved. Social and economic factors must always be considered for the adequate prevention and care of leprosy patients. Our model indicated no effects of the use of clofazimine or dapsone or BCG vaccination on the occurrence or severity of COVID-19. Public efforts, including vaccination, must be prioritized for vulnerable populations in leprosy-endemic countries to reduce the impact of pandemics on leprosy management.

## Supporting information

**S1 STROBE Checklist.**
(DOC)

**S1 Table. Raw data.**
(XLSX)

**S1 File. Additional analyses.**
(DOCX)

## Acknowledgments

We thank all of the professionals at the Hospital Universitário de Brasília, Brazil who were involved in the support of leprosy patients, especially during the COVID-19 pandemic.

## Author Contributions

**Conceptualization:** Selma Regina Penha Silva Cerqueira, Ciro Martins Gomes.

**Data curation:** Selma Regina Penha Silva Cerqueira, Patrícia Duarte Deps, Natanael Victor Furtunato Bezerra, Daniel Holanda Barroso, Ana Bárbara Sapienza Pinheiro, Simon M. Collin, Ciro Martins Gomes.

**Formal analysis:** Selma Regina Penha Silva Cerqueira, Patrícia Duarte Deps, Natanael Victor Furtunato Bezerra, Daniel Holanda Barroso, Simon M. Collin, Ciro Martins Gomes.

**Funding acquisition:** Ciro Martins Gomes.

**Investigation:** Selma Regina Penha Silva Cerqueira, Débora Vilela Cunha, Natanael Victor Furtunato Bezerra, Daniel Holanda Barroso, Taynah Alves Rocha Repsold, Patrícia Shu Kurizky, Ciro Martins Gomes.

**Methodology:** Daniel Holanda Barroso, Ana Bárbara Sapienza Pinheiro, Taynah Alves Rocha Repsold, Ciro Martins Gomes.

**Project administration:** Selma Regina Penha Silva Cerqueira, Ciro Martins Gomes.

**Resources:** Selma Regina Penha Silva Cerqueira, Débora Vilela Cunha, Gecilmara Salviato Pillegi, Ciro Martins Gomes.

**Software:** Selma Regina Penha Silva Cerqueira, Patrícia Duarte Deps, Simon M. Collin, Ciro Martins Gomes.

**Supervision:** Patrícia Shu Kurizky, Ciro Martins Gomes.

**Validation:** Gecilmara Salviato Pillegi, Patrícia Shu Kurizky, Ciro Martins Gomes.

**Visualization:** Selma Regina Penha Silva Cerqueira, Gecilmara Salviato Pillegi, Patrícia Shu Kurizky, Ciro Martins Gomes.

**Writing – original draft:** Selma Regina Penha Silva Cerqueira, Patrícia Shu Kurizky, Ciro Martins Gomes.

**Writing – review & editing:** Patrícia Duarte Deps, Gecilmara Salviato Pillegi, Simon M. Collin, Ciro Martins Gomes.

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
