## [Decision Letter · Decision Letter 0]

25 Jun 2021

Dear Dr. Gomes,

Thank you very much for submitting your manuscript "Leprosy during the COVID-19 pandemic as a model for assessing the effects of Bacillus Calmette–Guérin vaccine and clofazimine and dapsone therapy on SARS-CoV-2 infection: A prospective real-world cohort study." for consideration at PLOS Neglected Tropical Diseases. As with all papers reviewed by the journal, your manuscript was reviewed by members of the editorial board and by several independent reviewers. In light of the reviews (below this email), we would like to invite the resubmission of a significantly-revised version that takes into account the reviewers' comments. 

We cannot make any decision about publication until we have seen the revised manuscript and your response to the reviewers' comments. Your revised manuscript is also likely to be sent to reviewers for further evaluation.

Sincerely,

Vinicius M Fava, PhD

Associate Editor

Ana LTO Nascimento

Deputy Editor

Reviewer's Responses to Questions

**Key Review Criteria Required for Acceptance?**

**Methods**

-Are the objectives of the study clearly articulated with a clear testable hypothesis stated?

-Is the study design appropriate to address the stated objectives?

-Is the population clearly described and appropriate for the hypothesis being tested?

-Is the sample size sufficient to ensure adequate power to address the hypothesis being tested?

-Were correct statistical analysis used to support conclusions?

-Are there concerns about ethical or regulatory requirements being met?

Reviewer #1: The experimental design includes a prospective cohort study in which the main exposure was 2 previous BCG vaccinations, and the main outcome was COVID-19 infection. Also, secondary exposures were considered: BCG status (0, 1 or 2 doses) and leprosy status (patient, LHC or controls). This experimental design is quite interesting, but the complexity of the variables involved requires great caution when interpreting the data. In my view, the experimental design needs to be better explained. It is essential to clarify the following points: 

1. In the introduction, it was said that the aim of the study was to evaluate the effect of 2 doses of BCG on the occurrence of COVID-19. However, other approaches played a leading role in the methods and results, to test a lot of risk/protection factors to covid-19 outcome in leprosy patients. It is not consistent with the title of the paper initial proposed objective. The exploratory analysis and the results presented go far beyond what is presented in the title of the article, making it unsuitable for the work. I would like to suggest that the title as well as the excerpts relevant to the objective of the work be rewritten.

2. In the Methods section it was not clear what were the inclusion criteria for the Cohort study. It is essential to clarify these criteria for a better understanding of the experimental design. A flowchart detailing the experimental design and inclusion/exclusion criteria and criteria would be helpful. 

3. The strategy for defining the experimental size appeared to be for convenience, depending on the number of leprosy patients recruited in the period. It is important to clarify which method was used to calculate statistical power, and which parameters were used in the analysis.

4. The authors do not clearly define the group "active leprosy". What is the distribution regarding the clinical forms of leprosy? Were patients at any stage of treatment included?

5. The Brazilian population has a high degree of miscegenation. Population information about ancestry (or even race/ethnicity) was not available in the text. It would be important to consider these variables as possible confounders influencing the COVID-19 and leprosy outcomes, and if possible, reanalyze the data including this variable in the adjusted model.

6. How were defined the categories of these following variables: use of personal protective equipment (PPE) against SARS-CoV-2 infection (face mask, alcohol gel, frequent hand washing) and adoption of social distancing behaviours? It is important to explain better in the methodology.

7. For the statistical analysis the authors have used the R environment, but they mentioned only the R Studio. It is important to cite properly the R environment (R project) and all packages used.

Reviewer #2: The manuscript has a clear objective, is related to the theme and hypothesis to be tested, and has an appropriate research design to respond to the proposed objective. The population is described and appropriate for what is being analyzed by the researcher. I believe that the sample is adequate, however the presence of sample calculus was not observed. Statistical analysis are adequate to what is proposed in the study, however it is necessary to better describe the analysis of the data contained in the tables, adequate ethical precepts.

Reviewer #3: The methodology section is well defined and nicely described in the study

**Results**

-Does the analysis presented match the analysis plan?

-Are the results clearly and completely presented?

-Are the figures (Tables, Images) of sufficient quality for clarity?

Reviewer #1: 1. The authors mentioned that “409 patients were included in the study”. This can be confusing to the reader, since leprosy HHC and controls were also involved in the study. I suggest using another term, such as individuals.

2. The authors begin the results with the characterization of the population and then address secondary exposures. I would like to suggest that, once the main exposures of the work are clearly defined, that they come first in the order of results. 

3. In Brazil, the Ministry of Health recommends applying the second dose of BCG in household contacts of patients with leprosy. The results in Table 1 show this, where 55.55% of the leprosy HHC group received 2 doses of BCG. As expected, the control group had a much lower percentage of the 2 doses of BCG (15,71). In fact, this constitutes a huge bias for analyzes that assess the status of leprosy as a risk factor for COVID-19. If the authors made adjustments to the analyzes in order to correct for the influence of confounders, I believe that the uncorrected results should be omitted from the main text (they can be placed as supplementary) and only the adjusted RR and HR values should be presented. 

4. Table 3: It would be interesting to also show the p-values associated with the respective HR values (I suggest doing this for all tables). Some confidence interval values appear to be associated with borderline values of statistical significance.

5. The Table 5 presents possible risk factors (diabetes, obesity and thalidomide treatment, for example) for covid-19 in leprosy patients. It was not clear in the text that this approach was within the objectives of the work. From this cut-off, the sample size was 113 patients, reflecting some high confidence interval values. Do the authors consider this N sufficient to infer about these associations? The authors might consider placing this table as supplementary material.

Reviewer #2: The analysis follows the presented analysis plan. As for the results, I suggest that the information that does not appear in the tables be better described, but that need to be identified so that the data can be replicated by other studies and is better understood by the reader. In the relevant tables, I suggest adding, in the description, the presence of the patient's assessment in more than one variable. It would be interesting if the variables and their respective tests were identified in the footnotes of the tables for a better understanding of the reader.

Reviewer #3: Result section is nicely presented in the study.

**Conclusions**

-Are the conclusions supported by the data presented?

-Are the limitations of analysis clearly described?

-Do the authors discuss how these data can be helpful to advance our understanding of the topic under study?

-Is public health relevance addressed?

Reviewer #1: 1. In the discussion, it was hypothesized that leprosy would favor the risk of covid-19 due to immunosuppression, however TT leprosy patients, for example, may present a certain degree of cellular immune response. It would be essential to consider the complexity of the immune response to leprosy within this proposed hypothesis.

2. In which aspects related to COVID-19 infection in leprosy patients can the results contribute? The conclusion of the work focuses on the hypothesis of the influence of economic and social factors on vulnerability to COVID-19 in leprosy patients, but these data were not tested in the work. It would be interesting to conclude the article based on the results from the work, and its applicability.

Reviewer #2: The conclusion is addressed within the discussion topic, which is allowed by the journal, the considerations corroborate the study data. The existence of limitations was informed, however I felt the need for them to be better described.

The authors discuss the data clearly and objectively, favoring the understanding of the topic under study and addressing its relevance to public health.

Reviewer #3: Very good and given key message for SARS-CV-2 global Pandemic situation for leprosy control program.

**Editorial and Data Presentation Modifications?**

Reviewer #1: 1. I think that the Figure 1 is unnecessary. The Fig. 1 is only showing the time to measured outcomes, which could be explained in the text. 

2. Table 2: please correct: “Personal protection 125 125(89.29%)”

3. The Table 4 could be omitted from the main text, and included only in the supplementary material, since none of the results presented in it showed statistical significance.

4. The figures are not in good image quality. I indicate that the definition of the figures is improved.

5. Some sentences in the text are difficult to interpret. I suggest a good review of the English language.

Reviewer #2: It was observed in table 4 in the items clofazimine and thalidomide, a p value different from the analyzed calculation, which needs to be revised by the author, as the information in the footer referring to Fisher's exact test I ask it would be cellular or expected frequency?

Reviewer #3: No required

**Summary and General Comments**

Reviewer #1: Dear authors, 

The manuscript entitled “Leprosy during the COVID-19 pandemic as a model for assessing the effects of Bacillus Calmette–Guérin vaccine and clofazimine and dapsone therapy on SARS-CoV-2 infection: A prospective real-world cohort study” by Cerqueira et al., proposes to investigate the risk for COVID-19 development in leprosy patients, and to evaluate the effect of vaccination by BCG, in a sample from Brazil. The topic is extremely relevant to public health, considering the pandemic COVID-19 period, and the need for knowledge and contributions to contain the COVID-19 new cases and dissemination. The relationship between COVID-19 and leprosy arouses a lot of interest due mainly to the immunological complexity involving both diseases. The authors performed a 14-month prospective real-world cohort study and they investigated possible risk factors to COVID-19 outcome. They have included leprosy patients, household contacts and controls. The results showed that no characteristic related to leprosy or BCG vaccination acted as risk or protective factor against COVID-19. The methodology was well conducted; however, I have some observations and suggestions mainly regarding the experimental design and results. I believe that the writing of the manuscript needs a lot of improvement to clarify the rationale involved in the experimental design as well as the proposed objectives. I hope that the modifications suggested should contribute to improve the quality of the manuscript.

Reviewer #2: Study of interest, rich in important information for clinical practice, favors and expands knowledge about SARS-Cov-2 and its association with leprosy and with BCG vaccination, and regardless of the result, it is already an interesting and necessary proposal. This is the strength of the study, especially at this time we are experiencing. The weak point is related to the questions and suggestions already mentioned in the previous topics.

Reviewer #3: The manuscript Number PNTD-D-21-00714 entitled "Leprosy during the COVID-19 pandemic as a model for assessing the effects of Bacillus Calmette–Guérin vaccine and clofazimine and dapsone therapy on SARS-CoV-2 infection: A prospective real-world cohort study". This is very good study, which showing the risk of SARS-CoV-2 infection and its relation to BCG vaccine and clofazimine and dapsone treatment. The manuscript is well written and nicely presented. I would like the authors to consider the following points before it can be considered.

In table 2: Personal protection 125 125 (89.29%) ?

Secondary exposures of interest: use of PPE (Tables 1 e 2) ?

PLOS authors have the option to publish the peer review history of their article (what does this mean?). If published, this will include your full peer review and any attached files.

Reviewer #1: No

Reviewer #2: No

Reviewer #3: No
---

## [Editor Report · Decision Letter 1]

7 Jul 2021

Dear Dr. Gomes,

We are pleased to inform you that your manuscript 'The influence of leprosy-related clinical and epidemiological variables in the occurrence and severity of COVID-19: A prospective real-world cohort study.' has been provisionally accepted for publication in PLOS Neglected Tropical Diseases.

The editor complement the authors for addressing in details the issues raised by reviewers.

Best regards,

Vinicius M Fava, PhD

Associate Editor

Ana LTO Nascimento

Deputy Editor

---

## [Editor Report · Acceptance letter]

23 Jul 2021

Dear Dr. Gomes,

We are delighted to inform you that your manuscript, "The influence of leprosy-related clinical and epidemiological variables in the occurrence and severity of COVID-19: A prospective real-world cohort study.," has been formally accepted for publication in PLOS Neglected Tropical Diseases.

Best regards,

Shaden Kamhawi

co-Editor-in-Chief

Paul Brindley

co-Editor-in-Chief
